# Characterization of the Cell Wall Component through Thermogravimetric Analysis and Its Relationship with an Expansin-like Protein in *Deschampsia antarctica*

**DOI:** 10.3390/ijms23105741

**Published:** 2022-05-20

**Authors:** Luis Morales-Quintana, Daisy Tapia-Valdebenito, Ricardo I. Castro, Claudia Rabert, Giovanni Larama, Ana Gutiérrez, Patricio Ramos

**Affiliations:** 1Multidisciplinary Agroindustry Research Laboratory, Facultad Ciencias de la Salud, Instituto de Ciencias Biomédicas, Universidad Autónoma de Chile, Talca 3467987, Chile; luis.morales@uautonoma.cl; 2Facultad Ciencias de la Salud, Instituto de Ciencias Biomédicas, Universidad Autónoma de Chile, Temuco 4810101, Chile; daisy.tapia@uautonoma.cl (D.T.-V.); claudia.rabert@uautonoma.cl (C.R.); 3Multidisciplinary Agroindustry Research Laboratory, Carrera de Ingeniería en Construcción, Instituto de Ciencias Químicas Aplicadas, Universidad Autónoma de Chile, 5 Poniente 1670, Talca 3467987, Chile; ricardo.castro@uautonoma.cl; 4Agriaquaculture Nutritional Genomic Center, Las Heras 350, Temuco 4781158, Chile; giovanni.larama@cgna.cl; 5Facultad Ciencias de la Salud, Instituto de Ciencias Biomédicas, Universidad Autónoma de Chile, Santiago 8320000, Chile; 6Centro de Biotecnología de los Recursos Naturales (CenBio), Facultad de Ciencias Agrarias y Forestales, Universidad Católica del Maule, Talca 3480112, Chile; 7Plant Microorganism Interaction Laboratory, Centro del Secano, Facultad de Ciencias Agrarias y Forestales, Universidad Católica del Maule, Talca 3480112, Chile; 8Centro de Investigación de Estudios Avanzados del Maule (CIEAM), Vicerrectoría de Investigación y Postgrado, Universidad Católica del Maule, Talca 3480112, Chile

**Keywords:** cell wall, thermogravimetry analyses, molecular dynamic simulation, molecular modeling, expansin protein, antarctic plant

## Abstract

*Deschampsia antarctica* Desv. (Poaceae) is one of the two vascular plants that have colonized the Antarctic Peninsula, which is usually exposed to extreme environmental conditions. To support these conditions, the plant carries out modifications in its morphology and metabolism, such as modifications to the cell wall. Thus, we performed a comparative study of the changes in the physiological properties of the cell-wall-associated polysaccharide contents of aerial and root tissues of the *D. antarctica* via thermogravimetric analysis (TGA) combined with a computational approach. The result showed that the thermal stability was lower in aerial tissues with respect to the root samples, while the DTG curve describes four maximum peaks of degradation, which occurred between 282 and 358 °C. The carbohydrate polymers present in the cell wall have been depolymerized showing mainly cellulose and hemicellulose fragments. Additionally, a differentially expressed sequence encoding for an expansin-like (DaEXLA2), which is characterized by possessing cell wall remodeling function, was found in *D. antarctica*. To gain deep insight into a probable mechanism of action of the expansin protein identified, a comparative model of the structure was carried out. DaEXLA2 protein model displayed two domains with an open groove in the center. Finally, using a cell wall polymer component as a ligand, the protein–ligand interaction was evaluated by molecular dynamic (MD) simulation. The MD simulations showed that DaEXLA2 could interact with cellulose and XXXGXXXG polymers. Finally, the cell wall component description provides the basis for a model for understanding the changes in the cell wall polymers in response to extreme environmental conditions.

## 1. Introduction

The Antarctic region is one of the harshest ecosystems in the world. This continent is similar no other place on Earth and it is also considered a frozen desert, for this reason, in recent years, several studies have been conducted in different scientific areas, including studies with bacteria, protists, fungi, bryophytes, plants, and animals [1]. In general, the flora of Antarctica is restricted to a few species of lichens, bryophytes (mosses and liverworts), moss, micro- and macroalgae, and small vascular plants [2]. Only two angiosperms are present in Antarctic ecosystems: *Deschampsia antarctica* Desv. (Poaceae) and *Colobanthus quitensis* (Kunth) Bartl. (Caryophyllaceae) [3]. These two plants are described and widely studied by their capacity to be highly freezing tolerant [1,4]. Some articles have reported on the mechanisms by which these plants resist low temperatures [5,6,7]. Concerning these plants, *D. antarctica* has been the object of studies in many fields of biology: ecology, taxonomy, morphology, anatomy, reproduction, physiology, biochemistry, and molecular biology [5]. The harsh climatic conditions in Antarctica induce various mechanisms responsible for plant development and survival [5]. Some of the mechanisms induced by these extreme climatic conditions are the regulation of cell division and cell enlargement—two primary processes related with cell wall metabolism [8,9]. Addressing plant responses at the molecular and morphological level can give insights into the underlying adaptational mechanisms.

In this sense, the cell wall is a macro-molecular structure that regulates the growth and development of the plants’ cell and acts as the first barrier against different biotic and abiotic stresses [10,11,12]. The cell wall structure and metabolism have been investigated using different techniques in diverse plant species. An innovative method is the thermogravimetric analysis (TGA) that has been widely used in different investigations, and recently, we performed different studies on the changes in the physiological properties in the cell wall of the strawberry fruit [12,13,14]. Complementary, the expansins are proteins acting during the extensibility of the cell wall, and it has been described that the levels of transcripts coding for these proteins increase under stress conditions, suggesting an important role in the adaptability of plants to various adverse conditions [15,16,17]. The expansin proteins comprise non-enzymatic cell wall proteins that belong to a superfamily with four clades: the α-expansin and β-expansin family that are present in all land plants [18,19,20,21,22,23], and two other families named “like α-expansin” and “like β-expansin” proteins that have been found in plants [18,21,22,24], fungi [20], and bacteria [23]. Considerable evidence suggests that gene expression of expansins is induced by environmental stress factors such as heat, drought, heavy metals, salt, among others [25,26,27,28]. For example, the overexpression of an expansin gene isolated from *Oryza sativa* L. (named *OsEXPA10*) showed an increase during the plant growth stage [29]. In other more recent studies, five expansin-like B were identified as drought and heat stress-responsive genes in *Solanum tuberosum* L. [30]. A second study showed that expansin-like B1 overexpression (BrEXLB1) increases the drought tolerance and induces a higher rate of photosynthesis and root growth in the plants of *Brassica rapa* L. [31]. Recently, we showed that five expansin genes modulated by fungal endophytes in the Antarctic *C. quitensis* plants exposed to drought, increase their relative expression in response to hydric stress [17]. Despite this evidence showing the influence of expansin protein in the plant performance under drought stress, it is unknown if the expansins proteins can play a role in the cell wall remodeling in the *D. antarctica* plants.

In the present study, we investigated the percentage of degradation and thermal properties of the cell wall in *D. antarctica* plants in aerial tissues and roots through TGA. We related this percentage of degradation with the presence of a transcript of an expansin-like alpha, in both tissues, together with a structural approximation of the structure of its deduced protein and the protein–ligand interaction with the main monomers of the cell wall, looking for a relationship between the levels of the cell wall degradation and interaction of the expansin-like alpha protein with these ligands.

## 2. Result and Discussion

### 2.1. Thermogravimetric (TGA and DTG) Characterization

The structural organization, stability, and integrity of the polymers that constitute the cell wall in the aerial and root tissues from *D. antarctica* were evaluated through thermogravimetric analysis (TGA). Figure 1 shows a graphic with TGA curves, where the thermogram of each tissue can be observed. Moreover, the thermograms were divided into three regions, according to the changes in the curves pending, where two of these regions correspond to stages related directly to the cell wall composition [12]. The first stage between 50 and 180 °C shows how the samples mainly lose water, where it can be seen that there are no differences between the samples (Figure 1). The second stage or disassembly of the cell wall stage corresponds to the temperature range of 180 to 350 °C (Figure 1), where it is possible to observe the greatest differences between the samples. Region three with temperatures above 380 °C corresponds to the residue’s own cell and is not related to the cell wall [13]. The result shows that the aerial tissue has a high thermal stability (Figure 1); therefore, it is less stable and more prone to enzymatic degradation [32], and this difference can be related to its higher content in acetyl and hydroxyl groups [33].

In terms of the percentage of mass loss, in the first region of thermal degradation, no differences were observed between the samples, with percentage loss values to the aerial tissue and root dries of 3.07% and 2.75%, respectively (Table 1). For the second region of thermal degradation, the average difference was 2.13% between the two samples (Table 1), which can be explained by the differences in the polymeric composition of the two samples (mainly cellulose, hemicellulose, and other) [12,13].

Additionally, in Figure 2, is possible to observe more details of the change in the thermal stability by a deconvolution curve (DTG-TGA). The second region of Figure 1 (temperature between 180–380 °C) shows four peaks in Figure 2, even when the first peak (at 282 °C) is masked by the area of the second peak (312 °C). In this form, DTG-TGA (Figure 2) shows the shoulder formed by the decomposition of the first polymer type (to 282 °C of temperature) principally in the aerial tissues, which according to previous publications, correspond to hemicellulose polymer [34].

The hemicelluloses have a complex chemical structure that is rich in branch chains attached to the backbone through glycosidic linkages [35], and when the hemicellulose has a high concentration in the cell wall structure, it can relate to an increase in the crystallinity of the cellulose structure, and this form changes the thermal stability [36]. This stability can be measured by TGA in its three decompositions, physical loss of water, early pyrolysis (dehydration and cleavage) of the side chains of hemicellulose, and finally dehydration, decarboxylation, and decarbonylation [37].

Interestingly, an important difference was observed in the peak at 358 °C (Figure 2). At this temperature, Nam et al., (2020) described that is possible to observe the decomposition of cellulose in its crystalline form, with a high peak to the root tissue with respect to the aerial tissue [38]. The crystalline form is more stable than its amorphous form [39,40], this cellulose is a polymer with extended glycosidic linkages (1 → 4)-β and oriented with numerous regular inter- and intramolecular hydrogen bonds, and in the case of the roots, highly crystalline cellulose can be more resistant to enzymatic or protein degradation [41]. Figure 2 shows the region of temperature between 300 and 330 °C, where root and aerial samples did not exhibit differences in the peak size at 312 and 326 °C, respectively. This is an important point since it is possible to observe that the *D. antarctica* plant has a cell wall both in its root and in its aerial part, mostly composed of cellulose and hemicellulose, which should be more orderly in the case of the root (peak at 282 °C) than in aerial tissue. However, in both samples, they (hemicellulose and cellulose fragment) are the most important components for the composition of a cell wall that must protect the cell from stress conditions.

### 2.2. Differential Scanning Calorimetry (DSC) Analysis

The DSC thermograms of both samples (aerial and root tissues) are shown in Figure 3. The thermograms indicated the presence of a broad endothermic inflection at 97.17 °C and 120.46 °C, respectively. The most likely explanation is that it corresponds to the remanent water evaporation present in the samples. Additionally, the calorific requirement (or enthalpy normalized) result of the two samples can be calculated from Equation (1).
(1)$$\Delta H=\int_{T1}^{T2}Cp\;dT$$

In order to determine the stability of the system, the enthalpy normalized (calorific requirement J/g) was calculated (Figure 3). The result showed that the calorific heat requirement for the pyrolysis of the root samples is greater than aerial tissue with 1979.2 J g^−1^ and 1694.4 J g^−1^, respectively (see Figure 3). This would be explained by the higher water content of the sample, together with the presence of cellulose in its amorphous form present in the aerial tissues, over the crystalline cellulose of the root samples (Figure 1), and this is possible to conclude since the decomposition process required more energy in the root samples than in the aerial samples, due to the greater order of the polymers to be degraded in the root part.

### 2.3. Isolation of DaEXLA2 and Phylogenetic Analysis

Although the effect of expansin proteins on cell walls is known, different authors have related these proteins with the response of the plant to stress [17,30,42,43]. For example, Jones and McQueen-Mason (2004) report a high expression of the gene EXPA1 during drought stress compared to EXPA2 and EXPA3 in *Craterostigma plantagineum* Hochst [42]. Mateluna et al. (2017) showed that an expansin-like alpha and five other expansins are differentially expressed in the response to the inclination of radiata pine [43]. More recently, Chen et al. (2019) identified an organ-specific expression of several drought- and heat stress-responsive expansin-like B genes in *Solanum tuberosum* L. [30]. Additionally, the molecular mechanism underlying the plant response to abiotic stress such as saline stress is poorly understood in *D. antarctica*, and a partial expansin-like alpha sequence was found in the transcriptional database of *D. antarctica*. This transcriptional sequence was named expansin-like alpha 2 (DaEXLA2). Thus, the DaEXLA2 partial sequence was formed by 1084 bp. The deduced DaEXLA2 protein displays 236 aminoacidic residues.

A multiple sequence alignment analysis of DaEXLA2 with other expansin and expansin-like sequences showed a moderated level of amino acid conservation with over 41% with other expansins-like α (EXLA) proteins (Figure 4A). The sequence shows the characteristic domains I and II. In domain I, five cysteine (C) residues are present and one in domain II. In the domain II or cellulose-binding domain (CBD), tryptophan (W) residues are present.

A phylogenetic tree was performed with 36 other expansin amino acid sequences including expansins-like α and expansins-like β, α-expansins and β-expansins from Arabidopsis thaliana, and the other Antarctic vascular plant, *Colobanthus quitensis*. The analysis grouped the DaEXLA2 into the expansin-like clade (Figure 4B). 

### 2.4. DaEXLA2 Protein Model Obtention

The *D. antarctica* expansin-like α2 (DaEXLA2) structural models of the identified proteins were built based on the crystal structure of the Phlp1, a protein that was classified as pollen allergen, however, corresponds to β-expansin protein from Phleum pratense (PDB code: 1N10). The DaEXLA2 shares 32.99% of aminoacidic sequence identity with the 1N10. To obtain a stable structure, different evaluation methods were used to determine the best molecular conformation. Firstly, the root-mean-square deviation (RMSD), which is used for measuring differences between the backbones of proteins, was calculated. The RMSD values of 1N10 and DaEXLA2 were 2.08 Å, indicating that the DaEXLA2 structure is similar to the template. In fact, the DaEXLA2 showed a similar structure at the secondary and tertiary structure level, regardless of if it corresponds to a Like α-expansin and the template to β-expansin. Secondly, the stereochemical quality of the DaEXLA2 model was determined by PROCHECK program and the Ramachandran plot evaluation. In the model of DaEXLA2 protein, 100% of the residues were found in the favored regions (adding the most favorable regions, additionally allowed regions, and generously allowed regions). In this form, the model was validated at the stereochemical and structural levels (Table 2).

A ProSA analysis that evaluates the quality in energetical terms showed that the structural model is favorable because the obtained z-scores of the model were negative (Table 2). Finally, an analysis using Verify3D program showed favorable scores with 85% of the residues showing values over 1 score, and no residues were observed under 0 score values, indicating that no residues were modeled badly. Consequently, the final structure of the DaEXLA2 protein was accepted for subsequent analysis (Figure 5A). With respect to the structural characteristics of the DaEXLA2 model, the analysis showed that the model has a similar secondary structure and folding with respect to the template (Figure 5A).

The resulting structural model showed a similar folding that is characteristic of the other expansins, displaying two domains each with a small cavity, named open groove, that crosses both domains (Figure 5A). This corresponds to the typical characteristic of expansins previously described in the literature [17,43,44,45,46,47,48]. The open groove located in the central part of the DaEXLA2 structural model is the region through which the protein interacts with ligands [17,43,44,45,46,47,48]. In other proteins of the alpha-expansin family, it has been described as the area where the HFD motif is oriented, syndicated as the catalytic motif [18,19,20,21,22,23]. However, this motif has been described as absent in the expansin-like proteins [21,43], for the same reason it does not exist in DaEXLA2 or other Arabidopsis expansin-like proteins (Figure 4A). Other important residues for protein stability and protein–ligand interaction previously described [21,43,44] are three aromatic residues present in the open groove, which form pi–pi interaction with the sugar rings of the polymeric fractions of the different components of the cell wall [47]. In the case of DaEXLA2, these aromatic residues correspond to residues Y183, W203, and W215 (Figure 4A).

As mentioned above, the structure obtained for DaEXLA2 shared two domains: the first named “domain 1” or D1, commonly named the catalytic domain (Figure 5A). Additionally, a glycine loop is present in the region near the N-ter in the D1, the loop is described as important to protein–ligand interaction [44]. The structure of the expansin proteins is stabilized by three disulfide bonds that are formed by conserved six cysteine (C) residues, the structure has a D1 with a β-barrel fold which is formed by six β-sheets (Figure 5A). With respect to the “domain 2” or D2, it is named the cellulose-binding domain (CBD) and showed a β-sandwich fold that is formed by six β sheets and loops (Figure 5A). Additionally, both domains are linked through a short crossover loop that spans approximately 11 Å (Figure 5A). Independent of the expansin family member (Like, alpha or beta expansins), the open groove was described as important to the protein–ligand interaction [17,21,43,44,45,46,47,48,49], and it is shown using the red line in Figure 5B.

Mateluna et al. (2017) showed that the differences in the binding energy interaction obtained between six different expansin proteins differentially expressed in young pine (*Pinus radiata* D. Don) can be explained by changes in some residues that generate differences in electrostatic surface in the open groove region of two domains suggesting that these differences may help in substrate specificity of pine expansins [43]. For this reason, the electrostatic potential on the surface of DaEXLA2 protein model was analyzed (Figure 5C). The results showed that the front face of the protein is lightly electronegative in the open groove zone of the D1 and lightly electropositive in the open groove of the D2 (Figure 5C). These results were similar to those described for three different alpha expansins (PrEXPA1, PrEXPA2, PrEXPA4) from Radiata pine [43]. Interestingly, these results (those from DaEXLA2; Figure 5C) were in contrast to those described for PrEXLA1, a pine expansin-like alpha, which turned out to be more electropositive in the ligand-binding site (in the open groove) [43], indicating that this sub-family is probably more divergent in the interaction zone than the alpha expansins that have been described to date as mostly electronegative in the open groove [43,45,48].

### 2.5. Evaluation of the Protein–Ligand Interaction Mode

To evaluate the putative mode of protein–ligand interaction of the DaEXLA2 with different potential ligands considering the different structural members of the cell wall, a series of comprehensive docking was firstly performed (Table 3). As shown in Table 3, negative energies were obtained for the four ligands tested with the DaEXLA2 protein model, indicating a probable protein–ligand interaction [44]. Interestingly, the strongest binding was found between XXXGXXXG and the ligand; however, other ligands showed lower affinity energy value (Table 3). Additionally, the GAX ligand has an unfavorable orientation in the binding interaction with DaEXLA2, even when the energy is negative (Figure 6).

The substrate binding and the orientation mode inside the open groove in DaEXLA2 was analyzed (Figure 6). The ligands with the DaEXLA2 structure showed important differences in the orientation inside or outside of the open groove (Figure 6). The four ligands were oriented in the open groove, and especially in D1, close to an important loop previously described and named as the “glycine loop” [44]. This loop is part of the D1 but distant to the open groove, but it is very mobile and is oriented to approach the oriented ligands in the open groove (Figure 6). Independent of the ligand evaluated, with which the MD simulation was performed, the ligands have been oriented towards D1 in DaEXLA2 (Figure 6). However, the way in which they are oriented within D1 was different, and this could explain the differences in the energy levels shown in Table 3. This result differs from the other alpha expansins described, because previous studies described the cellulose fragment or Cellodextrin 8-mer as the best ligand [17,45,46,48], probably because this expansin (DaEXLA2) corresponds to an expansin-like protein.

Additionally, to obtain more reliable estimates of free binding energy than the values obtained from docking in Table 3, molecular mechanics/generalized Born surface area (MM-GBSA) calculations for each protein–ligand complex were performed (Table 4). The data confirmed that the affinity for XXXGXXXG is considerably higher than that for cellulose fragment or other XG ligands (Table 4). Other important point was that the major contributions to the ΔG_bind_ were the van der Waals forces and nonpolar solvation forces (Table 4). The total binding energy indicates that the complex between DaEXLA2 and XXFGXXFG is the most stable (–84.1 kcal mol^−1^). These data confirm the molecular dynamics simulation studies.

## 3. Material and Methods

### 3.1. Plant Samples

Plants of *D. antarctica* were collected in King George Island, South Shetland Islands. To obtain the transcriptomic data, plants were collected in areas near Collins Glacier shelter (62°10′05.6″ S, 58°51′06.2″ W), transported to the laboratory, propagated, and treated as described by Tapia-Valdebenito et al., 2016 [50]. Meanwhile for thermogravimetric analysis, plants of *D. antarctica* were collected on King George Island near Henryk Arctowski Polish Antarctic Station (62°09′34″ S; 58°28′19″ W) and transported by air to Temuco (Chile). In the laboratory, plants were cultivated using a soil/peat/vermiculite mixture (3:1:1), maintained, and propagated in a controlled environment plant growing chamber.

### 3.2. Thermogravimetric Analysis (TGA)

To determine the stability of the cell wall of *D. antarctica* plants, the thermogravimetric analysis (TGA) was used, and the root and aerial parts of the plant were first separated. Each plant part was homogenized with a mortar and pestle, and the maceration was then dried at 80 °C for 48 h according to Castro and Morales-Quintana (2019) [12]. The dry samples were ready for the following analyses. Dry samples of 5 mg from the three different plant groups were employed to evaluate the chemical characteristics of the degradation process by using a Discovery SDT-650 thermogravimetric analyzer (TA instrument), in which the samples were heated at a constant rate of 5 °C min^−1^ to temperatures between 50 °C to 500 °C using air as a reactive gas, and with a mass flow of 50 mL min^−1^. In addition, 50 mL min^−1^ of N2 was used as protective gas in the electronic balance.

Additionally, a differential scanning calorimetry (DSC) characterization was performed. The samples were introduced into the crucible (α-Al_2_O_3_), the analysis was performed from room temperature (23 °C) up to 550 °C by an SDT-Q600 simultaneous DSC–TGA instrument (TA Instruments, New Castle, DE, USA). Heating rate was conducted with an equilibration at 25 °C for one minute, followed by heating at 5 °C/min. The equipment was calibrated as per the manufacturer’s specifications using sapphire as the reference. For each analysis, 20 mg of the sample was used, the aerial tissue and roots carved from the fresh plants were kept at 5 °C and used 1 h after excision. The aerial tissue was cut into central longitudinal slices and was realized at 2 cm and the roots were cut into approximately 1 mm-thick slices, and for the root samples that were cut, a visual inspection was realized in order to detect possible defected regions. Transition enthalpy (ΔH expressed as J/g), onset temperature (To), peak temperature (Tp), and conclusion temperature (Tc) were determined by TRIOS TA-Instrument Thermal Analysis System Program.

### 3.3. Sequence Obtention and Analysis

Sequences of interest *D. antarctica* expansin-like alpha 2 (DaEXLA2) were identified in a transcriptome assembled from the sequencing of *D. antarctica* plants exposed to NaCl treatments according to Tapia-Valdebenito et al., 2016 [50]. The RNA was isolated from leaf samples using RNA-Solv^®^ purification reagent (Invitrogen, Waltham, MA, USA), followed by treatment with DNAse I (RQ1 RNase-Free DNase, Promega, Madison, WI, USA), and purified through columns using the E.Z.N.A Total RNA Kit I. The RNA samples were precipitated in 3 vol ethanol and 0.1 vol sodium acetate 3M, this solution was submitted to Macrogen Inc. (Seoul, Korea) for sequencing in an Illumina HiSeq 2000 platform for 100 cycles paired-end mode. The sequencing resulted in 225.6 million reads, which were assembled using a de novo approach with Trinity v2.8 [51], resulting in 75.760 unigenes with completeness of 77.3% assessed by BUSCO in eudicots_odb10 database [52]. The full-length transcript identified was deposited into the GenBank database (https://www.ncbi.nlm.nih.gov/genbank/, accessed on 22 March 2022) with ON042768 code. The sequence was translated to amino acids using the ExPASy Translate Tool (http://ca.expasy.org, accessed on 22 March 2022). The amino acid sequences were analyzed by a multi-sequence alignment using the ClustalW program associated with the BioEdit Sequence Alignment Editor v7.0 software [53]. Additionally, the phylogenetic tree construction was performed by neighbor joining (with 5000 bootstrap replicates) using MEGA X software [54]. The isoelectric point and molecular mass of the DaEXLA2 were predicted using the Compute pI/Mw tool (http://web.expasy.org/compute_pi/, accessed on 22 March 2022).

### 3.4. DaEXLA2 Structural Model Construction

MODELLER 9v18 software [55] was used to build the α-expansin-like structural model according to Morales-Quintana et al. (2011) [56]. The template search showed that the best crystal for structural mold corresponds to the crystal structure of the Phlp1, a protein that was classified as a major timothy grass pollen allergen, however, corresponds to β-expansin protein from Phleum pratense (PDB code: 1N10). Fifty structures were generated for the DaEXLA2 protein. The model quality was evaluated by PROCHECK [57] and ProSA-Web [58,59] programs. Additionally, the electrostatic surfaces of the DaEXLA2 were analyzed using the adaptive Poisson–Boltzmann solver (APBS) according to Mateluna et al., 2017 [43], and a potential scale from −2 to +2 KT/e was followed to use the APBS software within PyMol software [60].

### 3.5. Determination of the DaEXLA2–Ligand Interaction by Molecular Dynamics Simulation

As a first approach to the protein–ligand interaction, a molecular docking was carried out based on the methodology previously implemented in our laboratory described in Mateluna et al. (2017) [43] and used in Valenzuela-Riffo et al. (2018; 2019; 2020) [45,46,47]. Thus, docking studies were performed using the Autodock Vina program [61] to predict the putative binding interactions of the expansin proteins with four different octasaccharides as putative ligands: two xyloglucans (XG) (XXFGXXFG and XXXGXXXG polymers) as representative XG present in plants, a canonical glucuronoarabinoxylan (GAX), and one water-soluble cellulose fragment named cellodextrin 8-mer according to Valenzuela et al. (2019) [45] and Morales-Quintana et al. (2021) [17]. The ligands were built using GlyCAM software (http://glycam.ccrc.uga.edu/, accessed on 25 March 2022) and were obtained from a previous work carried out in our laboratory. Five independent docking runs were carried out. Additionally, a molecular dynamic (MD) simulation of each DaEXLA2 protein model was evaluated with the same four different octasaccharides as putative ligands. The MD simulation for the different expansin–ligand complexes was run during 100 ns, with 2 fs to the integration of the motion equations using the SCHRÖDINGER suite with the OPLS v2005 force field [62]. Every 50 ps of the trajectory, the data were collected and analyzed using the VMD software [63]. Finally, to estimate the relative binding free energies of the protein–ligand complexes, we employed the molecular mechanics/generalized Born surface area (MM-GBSA) methodology according to the protocol employed by Valenzuela-Riffo et al. (2019) [46]. We calculated the conformational entropy according to Vergara-Jaque et al. (2013) [64].

## 4. Conclusions

In conclusion, a sequence encoding for an expansin-like α from *Deschampsia antarctica* plants was identified, and the deduced protein was modeled and structurally characterized. In addition, the cell wall was evaluated by a novel technique describing the composition of this important cellular organelle in these plants. This research contributes deep insight into the biochemical and molecular knowledge of the stress tolerance mechanism used by plants that inhabit one of the most extreme environments on the planet. Additionally, the results provide information into the cell wall structural components, suggesting a specific affinity for the polymers in the cell wall, which results in the dynamic remodeling of cell walls in this Antarctic plant to face environmental stress.

## Figures and Tables

**Figure 1 ijms-23-05741-f001:**
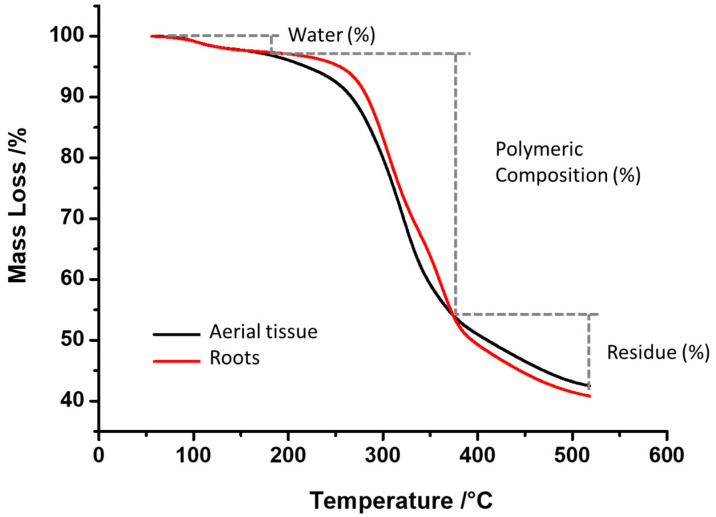
TGA curve. Thermogram derived from aerial and root tissues at temperatures between 50 and 550 °C.

**Figure 2 ijms-23-05741-f002:**
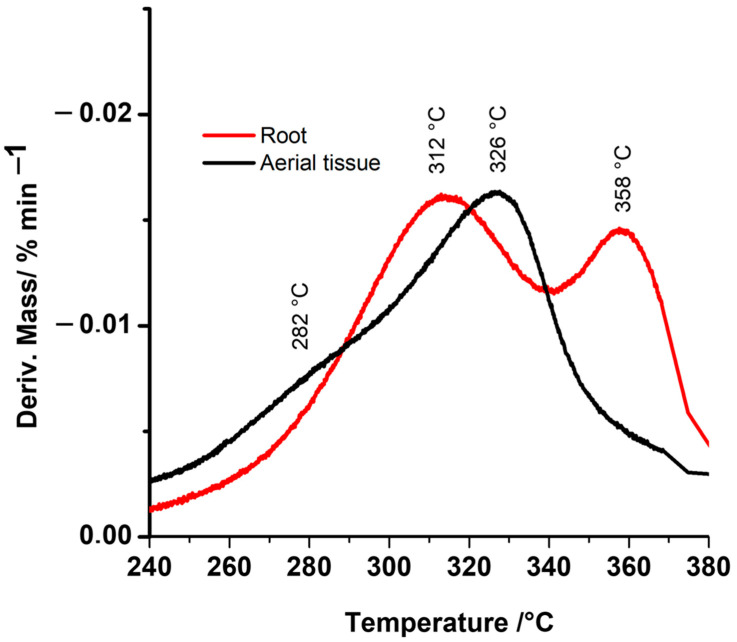
First derivative of the thermogram curves (DTG−TGA thermogram). DTG−TGA shows the maximum degradation temperatures of each cell wall component in the different tissues.

**Figure 3 ijms-23-05741-f003:**
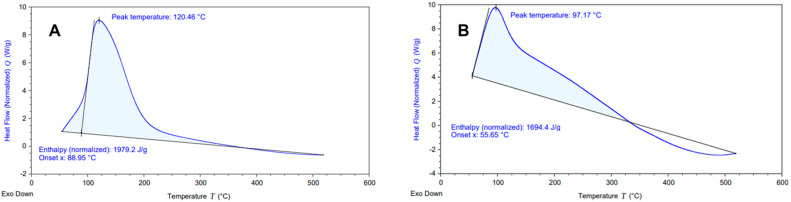
DSC thermograms of the root (**A**) and aerial (**B**) tissue of the *D. antarctica* plant.

**Figure 4 ijms-23-05741-f004:**
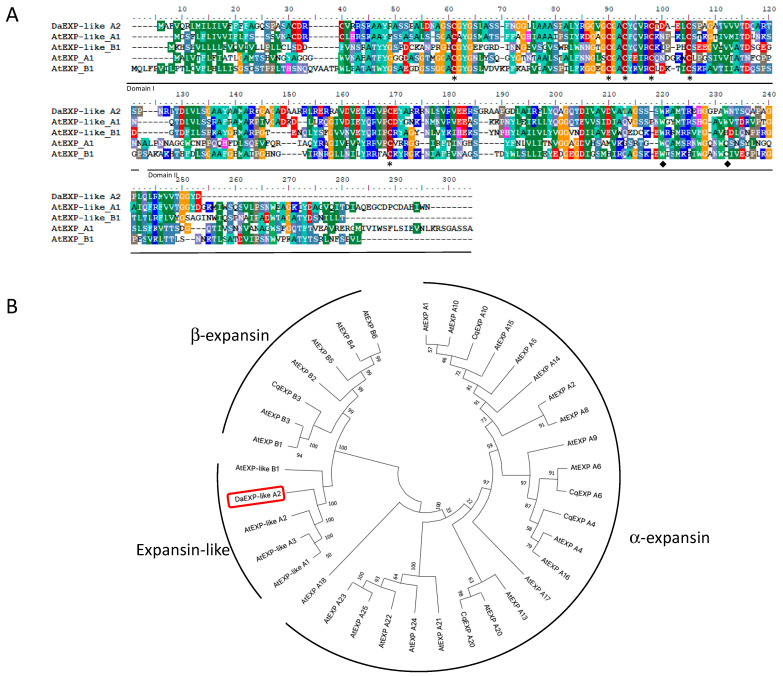
DaEXLA2 sequence analysis. (**A**) Multiple sequence alignment between different expansin proteins. Letters with the same color are identical residues or similar residues. Meanwhile, gaps are indicated by dashes. Asterisks show conserved Cys residues (C) of expansin proteins. Black diamonds indicate aromatic tryptophan residues (W) of expansins proteins. (**B**) Phylogenic analysis of the expansin from *D. antarctica* with the following different plant orthologs proteins: *Arabidopsis thaliana* AtEXPA1 (AEE34945), AtEXPA2 (AED90852), AtEXPA4 (AEC09708), AtEXPA5 (AEE77523), AtEXPA6 (AEC08194), AtEXPA8 (AEC09854), AtEXPA9 (AED90451), AtEXPA10 (AEE30732), AtEXPA13 (AEE73914), AtEXPA14 (AED96748), AtEXPA15 (AEC05663), AtEXPA16 (AEE79393), AtEXPA17 (AEE82054), AtEXPA18 (Q9LQ07), AtEXPA20 (NP_195534), AtEXPA21 (AED94413), AtEXPA22 (AED94414), AtEXPA23 (AED94415), AtEXPA24 (NP_198747), AtEXPA25 (AED94417), AtEXPB1 (AEC07066), AtEXPB2 (NP_564860), AtEXPB3 (AEE85459), AtEXPB4 (NP_182036), AtEXPB5 (NP_191616), AtEXPB6 (AEE34411), AtEXLA1 (AEE78096), AtEXLA2 (AEE86923), AtEXLA3 (AEE78095), AtEXLB1 (O23547). *Colobanthus quitensis* CqEXPA4 (MZ190884), CqEXPA6 (MZ190885), CqEXPA10 (MZ190886), CqEXPA20 (MZ190887), and CqEXPB3 (MZ190888).

**Figure 5 ijms-23-05741-f005:**
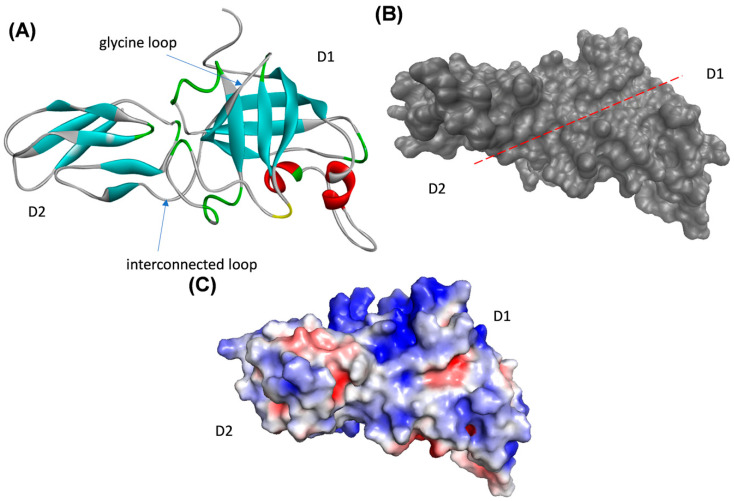
Structural model of DaEXLA2. (**A**) The protein model of DaEXLA2. In red the α-helices that form part of the domain I are shown and in cyan the β-sheets of the two domains are shown. Additionally, each domain was named D1 or D2 (domain I and II), and the interconnected loop is shown between both domains. (**B**) The protein surface and the open groove structure zone is shown using the red line. (**C**) Surface electrostatic potential of DaEXLA2 protein models; the electronegative zone is presented in red, the neutral zone in white, and the electropositive zone in blue.

**Figure 6 ijms-23-05741-f006:**
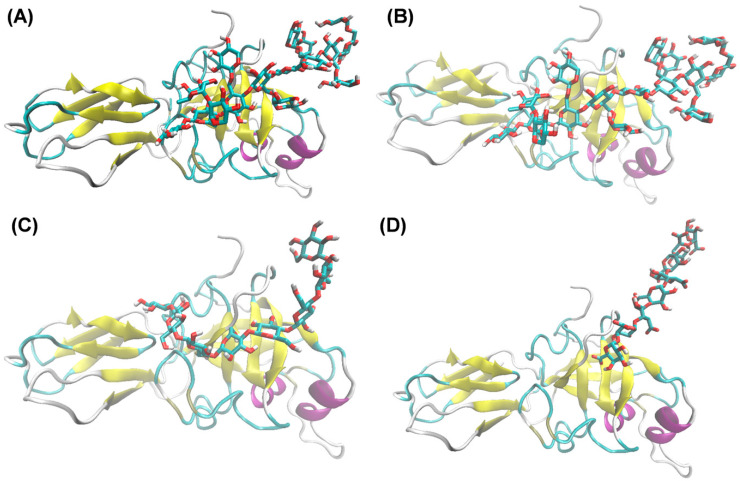
Protein–ligand interaction mode of DaEXLA2 with XXXGXXXG (**A**), XXFGXXFG (**B**), Cellodextrin 8-mer (**C**), and GAX (**D**).

**Table 1 ijms-23-05741-t001:** Percentage of the mass loss determination from *D. antarctica* by thermogravimetric analysis (TGA).

Fraction	Temperature °C	Average Percentage of Weight Loss (%)	
		Aerial Tissue	Root Tissue
1st degradation stage (Water loss)	50 to 180	3.07	2.75
2nd degradation stage (polymeric composition)	180 to 380	43.73	45.96

**Table 2 ijms-23-05741-t002:** Evaluation of DaEXLA2 structure after MD simulations and system equilibration.

Structure	Verify3D (Score)	ProSA (Z-Score)	PROCHECK			
			Core (%) ^a^	Allow (%) ^b^	Gener (%) ^c^	Disall (%) ^d^
DaEXLA2	85.2%	−4.09	80.2%	18.9%	0.8%	0.0%

^a^ Most favorable region; ^b^ Additionally allowed regions; ^c^ Generously allowed regions; ^d^ Disallowed regions.

**Table 3 ijms-23-05741-t003:** Affinity energy calculated mediating molecular docking methodology. Superscript letters indicate significant differences between the different ligands tested with each protein (Tukey HSD test, *p* < 0.05).

Protein	Ligand Name	Affinity Energy (kcal mol^−1^)
DaEXLA2	Cellodextrin 8-mer	−6.9 ^b^ ± 0.35
	XXXGXXXG	−8.2 ^a^ ± 0.11
	XXFGXXFG	−6.1 ^c^ ± 0.43
	GAX	−5.4 ^d^ ± 0.11

**Table 4 ijms-23-05741-t004:** MM-GBSA analysis for the interaction of FcEXPA1, FcEXPA2, and FcEXPA5 with cellulose and XXFGXXFG as the ligand.

Ligand	ΔH_vdWMM_ (kcal mol^−^^1^)	ΔH_elecMM_ (kcal mol^−^^1^)	ΔG_sol__–__pol_ (kcal mol^−^^1^)	ΔG_sol__–__npol_ (kcal mol^−^^1^)	ΔG_bind_ (kcal mol^−^^1^)
Cellodextrin 8-mer	−60.2	22.1	0.0	−41.0	–70.6 ± 0.7
XXXGXXXG	−65.7	27.3	0.0	−31.4	–69.8 ± 1.5
XXFGXXFG	−67.3	31.1	0.3	−48.2	–84.1 ± 0.8
GAX	−39.3	30.5	0.8	−41.2	–49.2 ± 3.5

ΔH_vdWMM_, van der Waals contributions; ΔH_elecMM_, electrostatic contribution; ΔG_sol__–pol_, ΔG_sol–npol_, contribution of solvation; ΔG_bind_, total binding energy.

## Data Availability

Not applicable.

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
