# Peer review of "Characterization of the Cell Wall Component through Thermogravimetric Analysis and Its Relationship with an Expansin-like Protein in Deschampsia antarctica"

_ijms, 2022, doi:10.3390/ijms23105741_

Round 1

Reviewer 1 Report

Comments to the author(s):

In this manuscript, the authors have identified sequence encoding for an expansin-like alpha protein obtained from the transcriptional database of Deschampsia antarctica vascular plant and modeled deduced protein, DaEXLA2, using homology comparative modeling. The Authors(s) have thoroughly checked their modeled protein structure using various well-known tools. Combined experimental, molecular docking, and MD simulation were performed to understand the molecular basis of the stress tolerance mechanism used by plants to cope with extreme environmental conditions. The work seems to be well conducted and described. However, a detailed analysis of cell wall polymer ligand interaction with DaEXLA2 model protein is lacking. Therefore, a careful comparison or discussion may be needed to explain the difference observed for DaEXLA2. I have several specific comments as follows:

  1. The authors state on page 7 that the resulting structural model showed a similar folding that is characteristic of the other expansins, displaying two domains each with a small cavity that crosses both domains (Fig. 5A). I would suggest adding a more detailed discussion especially, on the open groove that is important for protein-ligand interaction. Does the open groove segment have a different set of amino acids or conformation in the DaEXLA2 protein than the other variants? 
  2. Computational details are poorly written. Table 3 shows the affinity energy of the DaEXLA2 model with different potential ligands obtained through molecular docking, however, the protocol used for protein-ligand docking is missing. Since binding affinity and ligand conformation/pose are sensitive to search space, receptor conformation, rigid or flexible docking, and docking method/software, hence author should provide a detailed methodology used for DaEXLA2-ligand docking in the respective section. Authors should clearly state if they have adopted protocol from Valenzuela et al., (2019) or other references for molecular docking for the present study.
  3. The authors have discussed ligand’s affinity based on molecular docking/simulation conformation adopted within the D1 domain in protein. The amino acids that are critical in DaEXLA2 (in the open grove region) and are involved in polar and non-polar interactions with ligands are not discussed. Previously, cellodextrin 8-mer was identified as the best ligand for the α-expansins (Valenzuela-Riffo et al., 2018; 2019;Valenzuela-Riffo and Morales-Quintana, 2020; Morales-Quintana et al., 2021). However, why Cellodextrin 8-mer interacts differently in Like expansin A protein is not discussed at a molecular level. Through MD trajectory analysis or visualization of the most representative structures, the authors should evaluate the variation in the H-bonding interactions or dynamics for different ligands in the protein-ligand complexes. The analysis may give the insight to understand the molecular basis of the preference of different cell wall components in Like expansin proteins than that of similar family proteins.
  4. Molecular docking provides a very rough idea of the ligand’s affinity. The authors should consider including binding free energy computation using the Molecular mechanics-generalized Born surface area (MM-GBSA) or other efficient methods for a fair comparison
  5. In section 3.5 authors state that the molecular dynamic (MD) simulation of each DaEXLA2 protein model was evaluated with four different octasaccharides. The authors confirm if docking/MD simulations were performed with all fifty structures that were generated for the DaEXLA2 protein model.
  6. The authors should check for typographical errors throughout the manuscript. The following typographical error needs to be addressed: Table 3 instead of Table 2. (line 326) 

Author Response

Reviewer 1:

Comments to the author(s):

In this manuscript, the authors have identified sequence encoding for an expansin-like alpha protein obtained from the transcriptional database of Deschampsia antarctica vascular plant and modeled deduced protein, DaEXLA2, using homology comparative modeling. The Authors(s) have thoroughly checked their modeled protein structure using various well-known tools. Combined experimental, molecular docking, and MD simulation were performed to understand the molecular basis of the stress tolerance mechanism used by plants to cope with extreme environmental conditions. The work seems to be well conducted and described. However, a detailed analysis of cell wall polymer ligand interaction with DaEXLA2 model protein is lacking. Therefore, a careful comparison or discussion may be needed to explain the difference observed for DaEXLA2. I have several specific comments as follows:

R: Thank you very much for the comments. We have tried to include every point mentioned by the reviewer.

The authors state on page 7 that the resulting structural model showed a similar folding that is characteristic of the other expansins, displaying two domains each with a small cavity that crosses both domains (Fig. 5A). I would suggest adding a more detailed discussion especially, on the open groove that is important for protein-ligand interaction. Does the open groove segment have a different set of amino acids or conformation in the DaEXLA2 protein than the other variants?

R: More details have been included in the main text (line 275-290)

Computational details are poorly written. Table 3 shows the affinity energy of the DaEXLA2 model with different potential ligands obtained through molecular docking, however, the protocol used for protein-ligand docking is missing. Since binding affinity and ligand conformation/pose are sensitive to search space, receptor conformation, rigid or flexible docking, and docking method/software, hence author should provide a detailed methodology used for DaEXLA2-ligand docking in the respective section. Authors should clearly state if they have adopted protocol from Valenzuela et al., (2019) or other references for molecular docking for the present study.

R: Indeed, the reviewer is right and we apologize. It is a protocol commonly used by our group that by mistake we have not included the information in the document, but this has now been corrected (line 445-455)

The authors have discussed ligand’s affinity based on molecular docking/simulation conformation adopted within the D1 domain in protein. The amino acids that are critical in DaEXLA2 (in the open grove region) and are involved in polar and non-polar interactions with ligands are not discussed. Previously, cellodextrin 8-mer was identified as the best ligand for the α-expansins (Valenzuela-Riffo et al., 2018; 2019;Valenzuela-Riffo and Morales-Quintana, 2020; Morales-Quintana et al., 2021). However, why Cellodextrin 8-mer interacts differently in Like expansin A protein is not discussed at a molecular level. Through MD trajectory analysis or visualization of the most representative structures, the authors should evaluate the variation in the H-bonding interactions or dynamics for different ligands in the protein-ligand complexes. The analysis may give the insight to understand the molecular basis of the preference of different cell wall components in Like expansin proteins than that of similar family proteins.

Molecular docking provides a very rough idea of the ligand’s affinity. The authors should consider including binding free energy computation using the Molecular mechanics-generalized Born surface area (MM-GBSA) or other efficient methods for a fair comparison.

R: The MM-GBSA was included according to the reviewer suggestion (line 362-369) and table 4. We also included the methodology of the MM-GBSA in the materials and methods section (line 461-465)

In section 3.5 authors state that the molecular dynamic (MD) simulation of each DaEXLA2 protein model was evaluated with four different octasaccharides. The authors confirm if docking/MD simulations were performed with all fifty structures that were generated for the DaEXLA2 protein model.

R: We evaluated the 50 models and selected the best in energetic and structural terms, that structure was the one selected for all future analyses. In other words, all 50 models for protein-ligand interaction were not used. The method was re-written (445-455)

The authors should check for typographical errors throughout the manuscript. The following typographical error needs to be addressed: Table 3 instead of Table 2. (line 326)

R: We appreciate the reviewer's comment and the errors were corrected along the manuscript. 

Reviewer 2 Report

The authors of the manuscript described the thermal properties of the aerial and root tissues of the Deschampsia antarctica plant using two methods – TGA and DSC. Additionally, the authors identified a sequence encoding expansin-like A protein from this plant and the protein structure was modeled based on the crystal structure of beta-expansin protein Phlp1. The interaction of four oligosaccharides with DaEXLA2 was modeled by molecular dynamics simulation and binding affinities were determined for these ligands. The study is well designed and performed providing important insight into the stress tolerance mechanism used by D. antarctica plants. I recommend this manuscript for publication in this journal after minor revision.

-The DSC method is not fully described. There is unclear how the plant samples were prepared for this experiment. Also the applied heating rate was not defined. The heating rate may influence the conversion of the cell wall components, therefore should be indicated.

-The authors could analyze cellulose, hemicellulose and lignin quantity of root and aerial parts of the plants by biochemical assays to compare the data with the thermogravimetric data.

-Authors also need check grammar mistakes and typo errors. There is inconsistent usage of several abbreviations in the manuscript: expansin-like alpha protein, alpha-like expansin, alpha expansin like, Like α- expansin, therefore, this makes it difficult to read the article.

  1. Line 39. “using cell wall polymer as ligand” should be changed to “using cell wall polymer components as ligands”
  2. Line 397 cellodextrin 8-mer should be named as cellulose fragment, but not as cellulose molecule.

Author Response

Reviewer 2:

Comments to the author(s):

The authors of the manuscript described the thermal properties of the aerial and root tissues of the Deschampsia antarctica plant using two methods – TGA and DSC. Additionally, the authors identified a sequence encoding expansin-like A protein from this plant and the protein structure was modeled based on the crystal structure of beta-expansin protein Phlp1. The interaction of four oligosaccharides with DaEXLA2 was modeled by molecular dynamics simulation and binding affinities were determined for these ligands. The study is well designed and performed providing important insight into the stress tolerance mechanism used by D. antarctica plants. I recommend this manuscript for publication in this journal after minor revision.

R: Thank you very much for the comments. We have taken into account all comments and tried to include every point mentioned by the reviewer.

-The DSC method is not fully described. There is unclear how the plant samples were prepared for this experiment. Also, the applied heating rate was not defined. The heating rate may influence the conversion of the cell wall components, therefore should be indicated.

R: As the reviewer suggested, the DSC method was described in more detail (line 401-410).

-The authors could analyze cellulose, hemicellulose and lignin quantity of root and aerial parts of the plants by biochemical assays to compare the data with the thermogravimetric data.

R: With respect to the inclusion of cell wall fractionation analysis, these studies would undoubtedly be an interesting view that would support us in the understanding of the phenomenon. However, the orientation of the study rather than seeking a detailed analysis of each component of the plant cell wall (which has been widely studied in the past by various authors), was to improve our understanding of the phenomenon of the drought defense process and therefore the cell wall was studied as a macro-complex that makes up a whole of the cell wall by TGA.

-Authors also need check grammar mistakes and typo errors. There is inconsistent usage of several abbreviations in the manuscript: expansin-like alpha protein, alpha-like expansin, alpha expansin like, Like α- expansin, therefore, this makes it difficult to read the article.

R: We appreciate the reviewer’s comment. The manuscript was checked and corrected as suggested.

Line 39. “using cell wall polymer as ligand” should be changed to “using cell wall polymer components as ligands”

R: The sentence was changed (line 40).

Line 397 cellodextrin 8-mer should be named as cellulose fragment, but not as cellulose molecule.

R: The sentence was changed (359).

Round 2

Reviewer 1 Report

The authors have revised the manuscript appropriately. Good Luck!

In line 356 the authors have used the "MDS" abbreviation. The authors should mention what "MDS" stands for?